# Metasurface Holography with Multiplexing and Reconfigurability

**DOI:** 10.3390/nano14010066

**Published:** 2023-12-26

**Authors:** Yijun Zou, Hui Jin, Rongrong Zhu, Ting Zhang

**Affiliations:** 1Interdisciplinary Center for Quantum Information, State Key Laboratory of Modern Optical Instrumentation, ZJU-Hangzhou Global Scientific and Technological Innovation Center, Zhejiang University, Hangzhou 310027, China; 12331098@zju.edu.cn (Y.Z.); 12231112@zju.edu.cn (H.J.); rorozhu@zju.edu.cn (R.Z.); 2School of Information and Electrical Engineering, Zhejiang University City College, Hangzhou 310015, China; 3College of Information Science & Electronic Engineering, Zhejiang Provincial Key Laboratory of Information Processing, Communication and Networking (IPCN), Zhejiang University, Hangzhou 310027, China

**Keywords:** metasurface, holography, multiplexing, reconfigurability, multifunctional metadevices

## Abstract

Metasurface holography offers significant advantages, including a broad field of view, minimal noise, and high imaging quality, making it valuable across various optical domains such as 3D displays, VR, and color displays. However, most passive pure-structured metasurface holographic devices face a limitation: once fabricated, as their functionality remains fixed. In recent developments, the introduction of multiplexed and reconfigurable metasurfaces breaks this limitation. Here, the comprehensive progress in holography from single metasurfaces to multiplexed and reconfigurable metasurfaces is reviewed. First, single metasurface holography is briefly introduced. Second, the latest progress in angular momentum multiplexed metasurface holography, including basic characteristics, design strategies, and diverse applications, is discussed. Next, a detailed overview of wavelength-sensitive, angle-sensitive, and polarization-controlled holograms is considered. The recent progress in reconfigurable metasurface holography based on lumped elements is highlighted. Its instant on-site programmability combined with machine learning provides the possibility of realizing movie-like dynamic holographic displays. Finally, we briefly summarize this rapidly growing area of research, proposing future directions and potential applications.

## 1. Introduction

Holography, initially conceived by Dennis Gabor in 1948 [1], serves as a technique for capturing and reconstructing full-wave information from objects. However, conventional optical holography requires a complex photographic process to record the interferogram pattern formed by the scattered light from an object and a coherent beam. This makes optical holography susceptible to environmental factors such as temperature, humidity, and light, leading to compromised imaging quality. In response to this limitation, Brown and Lohman introduced the concept of computer-generated holography (CGH) in 1966 [2]. In CGH, wavefront information at the hologram plane is numerically calculated using diffraction theory, simplifying the recording process through computer programming [3,4]. Compared with traditional optical holography, CGH not only facilitates the reconstruction of virtual objects but also enhances imaging quality through optimization algorithms, which greatly increases the degree of design freedom. Moreover, the integration of CGH with digital light-field modulators, including spatial light modulators (SLM) [5] and digital micromirror devices (DMD) [6,7] equipped with dynamic light manipulation capabilities, enables the realization of multifunctional holography [8,9]. However, materials like phase-modulated materials with a finite refractive index accumulate sufficient phase changes only when the light propagates over a distance much larger than the wavelength. Therefore, the size and thickness of the optical elements used to construct the phase hologram become significantly larger than the wavelength, which lead to high-order diffraction, low imaging efficiency, and limited resolution of the hologram [10]. Therefore, identifying superior modulation materials as alternatives has become a pressing concern in the field of optical holography.

In recent years, significant progress in nanofabrication technology holds the potential to revolutionize holography. Metasurfaces [11,12,13,14,15,16,17,18,19], as two-dimensional (2D) forms of metamaterials, typically comprise arrays of subwavelength planar optical elements with spatial geometric variations. In comparison to digital light-field modulators and metamaterials, metasurfaces not only have the powerful ability to modulate optical properties on the subwavelength scale but also offers advantages such as low absorption loss, ultra-thinness, and small pixel size. Metasurfaces provide a new perspective for the design of various optical devices, such as orbital angular momentum devices [20,21,22,23,24], cloak devices [25,26,27,28,29,30,31], and ultra-thin planar lenses [32,33,34] and spectroscopes [35,36]. A cutting-edge application of nanotechnology is the combination of metasurface and CGH, which composes metasurfaces by mapping CGH-generated holograms based on the local scattering properties of predesigned nanostructures to realize holography. Metasurface holography offers several advantages over previous holographic implementations, such as higher spatial resolution, low noise, larger frequency bandwidth, and the elimination of unwanted diffraction levels [37,38]. Consequently, metasurfaces are considered promising devices for applications such as display [39,40], imaging [41,42], encryption [43,44], etc.

Since great achievements have been made to realize customized single metasurfaces, more and more researchers have focused on the integrated design of metasurfaces that can deal with concurrent holographic tasks. Multiplexing is a concept in telecommunication and combines multiple signals into one signal. In metasurface holography, multiplexing refers to the integration of different holographic displays into a single metasurface and switching them by different properties of light (Figure 1). By pre-considering the classes of degrees of freedom, multiplexed holograms can be obtained using the CGH combined with multi-objective optimization algorithm. When encoding multiplexed holograms into metasurfaces, segmented and interleaved configurations are most commonly used. On the other hand, reconfigurable metasurface do not aim at the nature of light but rather affect the local optical response through changing the structural parameters, which provides an alternative new method for realizing the integration of different holographic displays.

In this paper, we present a comprehensive overview of the progression from single metasurface holography to multiplexed and reconfigurable metasurface holography. In Section 2, we introduced phase-only, amplitude-only, and complex amplitude holography. In Section 3, we focus on orbital angular momentum multiplexed metasurface holography, including the theoretical design and application. The subsequent discussion extends to angle-selective, wavelength-selective, and polarization-selective multiplexed metasurface holography. In the next section, we offer an in-depth review of the design theory and applications of reconfigurable metasurface holography, emphasizing the integration of electrically tunable metasurfaces with machine learning techniques. In the last section, we provide an overview of the future research perspectives and the challenges that lie ahead in the realm of metasurface holography. 

## 2. Single Metasurface Holography

Metasurface holography is categorized into three types based on the distinction between the metasurface and electromagnetic modulation component in computer-generated holography (CGH): phase-only metasurface holography, amplitude-only metasurface holography, and complex amplitude metasurface holography.

### 2.1. Phase-Only Metasurface Holography

The wavefront profiles of phase-only metasurface holograms can be generated by the Gerchberg–Saxton (GS) or point source algorithm. These algorithms simulate the diffuse reflection of objects by incorporating random phase masks to achieve a uniform amplitude distribution. Among various types of metasurfaces, geometric metasurfaces based on the Pancharatnam–Berry (PB) principle exhibit excellent phase control capabilities. The abrupt phases related to the direction of spatial change are frequency-independent (dispersion-free) and completely depend on the orientation angle of the antenna [45,46]. Huang et al. [47] demonstrated a metasurface hologram by the geometric phase principle, as shown in Figure 2a. For circularly polarized incident light, the metasurface hologram can achieve the expected phase distribution in the orthogonal circularly polarized output light, and a 3D reconstructed image with resolution, a large field of view, and no multi-order diffraction and twinning is displayed. However, the inherent ohmic loss in plasmonic materials will cause the inefficient diffraction of the proposed visible wavelength hologram. To solve this problem, Zheng et al. [48] demonstrated a reflective geometrical metasurface hologram based on grounded metal planes. As shown in Figure 2b, the hologram has a diffraction efficiency of 80% at 825 nm and ultra-high bandwidth from 630 nm to 1050 nm. Notably, these geometric metasurface holograms can withstand up to 10% fabrication defects, including shape deformation and phase noise, which greatly reduces fabrication difficulties. In addition to the above phase modulation of the main mode, the phase of the second harmonic (SH) can also be nonlinearly manipulated. In 2018, Ghirardini et al. [49] demonstrated the shaping of the SH radiation pattern from a single AlGaAs nanodisk antenna using coplanar holographic gratings. The use of such gratings allows increasing the SH power collection efficiency by two orders of magnitude with respect to an isolated antenna. Such reconstruction of the nonlinear emission from nanoscale antennas represents the first step toward the application of all-dielectric nanostructures for nonlinear holography. In addition, phase-only metasurfaces are also widely used in metalens [32,34], which can replace traditional refractive lenses to improve the compactness and efficiency of holographic imaging systems. In 2018, Chen et al. [50] showed that, by judicious design of nanofins on a surface, it is possible to simultaneously control the phase, group delay, and group delay dispersion of light, thereby achieving a transmissive achromatic metalens with a large bandwidth. Chen et al. demonstrated diffraction-limited achromatic focusing and achromatic imaging from 470 to 670 nm.

### 2.2. Amplitude-Only Metasurface Holography

Amplitude, as an optical field component, also can be regarded as one of the degrees of freedom in the design of metasurface holography. In amplitude-only metasurface holography, the local transmission or reflection amplitude of each meta-atom can be quantitatively divided into different levels. The simplest and common strategy is to assume only two amplitude values: 0 and 1. Butt et al. [51] used vertically aligned arrays of multi-walled carbon nanotubes as pixels to realize a binary amplitude hologram, as shown in Figure 2c. However, the binary hologram mentioned above suffers from the twin image problem. To solve this problem, Huang et al. [52] analyzed the diffraction field of a large number of subwavelength photon sieves and then used the genetic algorithm (GA) for optimization to achieve a uniform, twin-free, and highly efficient binary amplitude hologram, as shown in Figure 2d. However, the information storage capacity of binary holograms is inefficient. To solve this problem, Walther et al. [53] tuned the transmission coefficient through the microscopic description of nanoholes in metal films of different sizes and demonstrated multistage amplitude holography at two wavelengths, which can be approximated by holes perforated in a metal film as components of a dipole emitter.

**Figure 2 nanomaterials-14-00066-f002:**
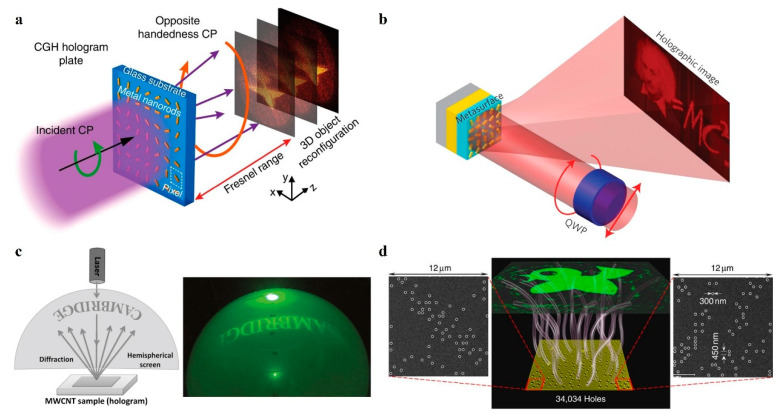
Phase-only and amplitude-only metasurface holography. (**a**) Schematic of three-dimensional optical holography using a plasmonic metasurface [47]. (**b**) Schematic of metasurface holograms reaching 80% efficiency [48]. (**c**) Schematic of carbon nanotube-based high-resolution holograms [51]. (**d**) Schematic of ultra-high-capacity nonperiodic photon sieves operating in visible light [52].

### 2.3. Complex Amplitude Metasurface Holography 

In fact, in order to reconstruct high-quality images without losing any information, the realization of an arbitrary complex wavefront requires the simultaneous modulation of phase and amplitude. Based on the Babinet principle, Shalaev et al. [37] proposed a V-shaped nanoantenna with two levels of amplitude and eight levels of phase modulation and realized a reconstruction image with high resolution and low noise in the visible range, as shown in Figure 3a. However, the bandwidth for a plasmon resonance-tuned metasurface based on the symmetric and antisymmetric resonance modes is very limited. Wang et al. [54] used a CSRR with broadband characteristics as the basic meta-atom and simultaneously manipulated the amplitude and phase of the outgoing orthogonally polarized linear wave by changing the geometrical parameters (radius r, split angle α, and orientation angle θ). The complex amplitude hologram with five levels of amplitude modulation and eight levels of phase modulation is shown in Figure 3b.

Another strategy for realizing complex amplitude modulation is to expand the geometrical metasurface. Lee et al. [55] proposed an X-shaped meta-atom that could provide two independent modes of PB phase superposition to independently and completely control the amplitude and phase distributions at a subwavelength spatial resolution. As shown in Figure 3c, the experimental demonstration at visible wavelengths was realized based on this meta-atom. In addition, Overvig et al. [56] proposed a dielectric metasurface composed of meta-atoms with different forms of birefringence and rotation angles. As shown in Figure 3d, metasurfaces control the amplitude by structurally birefringent meta-atoms changing the conversion efficiency of one-handed circularly polarized light to backhandedly polarized circularly polarized light and the phase by the in-plane orientation of the meta-atoms.

## 3. Multiplexed Metasurface Holography

With the potential of huge spatial bandwidth product and information capabilities, metasurfaces are very suitable for developing multiplexing techniques based on different optical properties. In this section, we present recent advances in multiplexed metasurface holography, including OAM-multiplexed, wavelength-multiplexed, angle-of-incidence-multiplexed, and polarization-multiplexed metasurface holography.

### 3.1. Orbital Angular Momentum Multiplexed Metasurface Holography

The orbital angular momentum (OAM) is of great interest as one of the fundamental physical properties. Vortex beams with OAM have a donut-shaped intensity distribution and exhibit a helical phase factor $e^{il\varphi }$, where l denotes the topological charge number, and φ denotes the azimuthal angle. Because of the orthogonality between different OAM modes, it is considered to be the perfect approach to realize optical multiplexing, which plays an important role in applications such as optical communication [57,58,59], stimulated emission loss microscopy [60,61], and optical tweezers [62,63]. Recently, there have been a number of multiplexed metasurface holography techniques with OAM as the degree of freedom proposed. In 2019, Ren et al. [64] demonstrated OAM metasurface holography with GaN nanopillars. As shown in Figure 4a, three kinds of metasurface holograms with discrete spatial frequency distributions have been proposed, including OAM-conserving, selective, and multiplexed metasurface holograms, where OAM beams with different topological charges can reconstruct different character images. 

In addition, orthogonality makes OAM naturally have huge advantages in data encryption. In 2020, Zhou et al. [43] combined OAM and polarization selectivity, proposing a technique for holographic information encryption and image generation using an all-media birefringent metasurface. Interestingly, this method provides additional degrees of freedom for erasing and modifying the holographic image, similar to the always-known stimulated emission depletion (STED) technique in microscopy [65,66]. Furthermore, the number of topological charges between different OAM beams is infinite and thus has tremendous potential for data storage. In 2021, Ren et al. [67] demonstrated a momentum-space ultra-high-dimensional large-scale OAM multiplexed holography on the basis of a complex amplitude metasurface, as shown in Figure 4c. Vortex beams that range from −50 to 50 OAM modes are sequentially incident on the metasurface hologram in order to solve the orthogonal image framing problem of OAM, and two different holographic videos will be reconstructed simultaneously in the momentum space. In addition, OAM can also solve the coupling problem of nonlinear waves. In 2021, Fang et al. [68] demonstrated a high-dimensional OAM multiplexed nonlinear holography. As shown in Figure 4d, through combining the class II second harmonic generation process [69], different OAM holographic images in the fundamental and second harmonics can be reconstructed independently in the spatial frequency domain. 

**Figure 4 nanomaterials-14-00066-f004:**
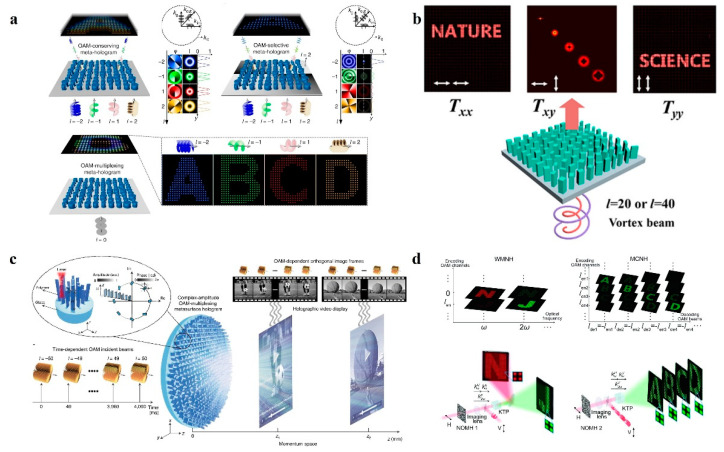
OAM multiplexed metasurface holography. (**a**) Schematic of metasurface orbital angular momentum holography by utilizing the strong orbital angular momentum selectivity offered by meta-holograms consisting of GaN nanopillars with discrete spatial frequency distributions [64]. (**b**) Schematic of OAM multiplexing in different polarization channels using a birefringent metasurface for holographic encryption [43]. (**c**) Schematic of ultra-high-dimensional OAM-multiplexed holography based on a large-scale complex-amplitude OAM-multiplexed metasurface hologram [67]. (**d**) Schematic of OAM multiplexing nonlinear holography [68].

### 3.2. Wavelength Multiplexed (Colorful) Metasurface Holography

Conventional optical holography usually works at single frequency point because of the limitation of the diffraction principle. However, the realization of specific optical functions at different wavelengths is a fundamental requirement for integrated photonics, such as colorful holographic displays. Currently, metasurfaces with interleaved designs become an effective method for wavelength-multiplexed and multifunctional meta-device designs. In 2015, Huang et al. [70] presented an interleaved nanoblocks structure that consisted of four subunits to achieve independent-phase modulation for the linear polarization of red, green, and blue. A colorful metasurface holography was achieved, as shown in Figure 5a. However, the orientation directions of all the nanoblocks above are the same, which makes the device only obtain a phase difference of 0~π. In order to break this limitation, Wang et al. [71] changed the orientation angle among the nanoblocks to achieve full-phase modulation under circular polarization, and the corresponding achromatic and high-dispersion colorful holograms are shown in Figure 5b. 

In addition, the interleaved metasurfaces can provide some control of polarization. As shown in Figure 5c, the meta-atom consisted of two interlaced nanoblocks. Each nanoblock could approximately independently control the phase of a specific wavelength and polarization of the beam [39]. As shown in Figure 5d, the holographic images of “chameleon” in LCP green light and RCP red light are displayed at the same time, and the color of “chameleon” will be changed by control of the polarization of the incident wave. The realization of a full-color gamut holographic display has always been a human dream, but the current color holography mainly focuses on hue and saturation, with little exploration of luminance. Bao et al. [40] proposed a dielectric metasurface made of crystal silicon nanoblocks. The meta-atom not only achieved a customizable coverage of the three primary colors but also enabled intensity control. The color gamut of holographic images was extended from 2D to 3D HSB space. Moreover, as shown in Figure 5f, a single-layer silicon metasurface can simultaneously display arbitrary HSB color nano-printed and full-color holographic images. 

### 3.3. Angular Multiplexed Metasurface Holography

In metasurface holography, plane waves (Gaussian beam excitation) are the most common type. The plane wave imposes a constant (normal incidence) or linear gradient (titled incidence) phase on the metasurface and is modulated into the desired wavefront. Typically, when the angle of incident deviates from the design, the holographic image is shifted or distorted, as shown in Figure 6a. Currently, there are some strategies to break this limitation. In 2017, Kamali et al. [72] demonstrated an angle-multiplexed metasurface holography composed of U-shaped dielectric resonators. As shown in Figure 6b, it can excite symmetric and antisymmetric resonance modes at different incidence angles, and the tremendous difference between the two modes exhibits the potential for independent phase modulation. The proposed angle-multiplexed metasurface hologram can encode different holographical images under $0^{\circ }$ and $30^{\circ }$ incidence angles with TE polarization, as shown in Figure 5c. Similar to this principle, Shuai et al. [44] further proposed a Fabry–Perot (FP) [73] resonator meta-atom. The discrepancy between the critical resonance lengths of the surface plasmon and MIM nanocavity for different illumination angles allows modulating the phase and amplitude at the same time. As shown in Figure 6d, this method enables independent encryption that displays near-field microscopic images (3D dice) at $\theta_{1}$ and far-field holographic images (K or Q) at $\theta_{2}$. 

In addition, the angular multiplexed technique is a suitable method for independent multichannel wavefront control. In 2020, Zhang et al. [74] combined wraparound-phase holograms with spatial multiplexing to record four phase profiles in a single metasurface hologram. As shown in Figure 6e,f, four different images can be generated independently with high fidelity, depending on the incidence angle. The wavefront control scheme can be applied not only to metasurface holographic multiplexing but also extended to multifunctional planar optics and wearable devices.

### 3.4. Polarization Multiplexing Metasurface Holography

As a transverse wave, an electromagnetic wave has a polarization property. Traditional CGH devices are either polarization-insensitive (diffractive optical elements) [75,76] or can only operate in specific polarization states (liquid crystal) [77]. Metasurface holograms consisting of anisotropic subwavelength structures can provide the ability to respond differently, depending on the polarization state. This property makes them suitable for polarization multiplexed holography. In 2020, Guan et al. [78] achieved two different information channels by manipulating the transmitted cross-polarized and co-polarized components of a 1-bit encoded metasurface at linearly polarized incidence. The orientation of the double-layer open ring (SR) aperture of the meta-atom was specifically designed to be 45° or 135° to achieve the same multiplexing functionality for both x-polarized and y-polarized incidences. A proof-of-concept experiment is demonstrated in Figure 7a, as the proposed coded metasurface holograms could project two separate holographic images at the same time without altering the incidence state and avoided the crosstalk between the different channels. In addition to linear polarization states, circularly polarization states can also be considered as a degree of freedom for metasurface holograms. Muller et al. [79] combined the geometric and propagating phases to achieve two independent and arbitrary phase distributions for any pair of orthogonal polarization states (linear, elliptical, or circular). Muller et al. demonstrated chiral metasurface holography for left- and right-handed circularly polarization states, respectively, as shown in Figure 7b. Circularly polarization (CP) modulation based on the geometric phase (Pancharatnam–Berry (PB)) [80,81] has been widely explored for metasurface engineering. However, the inherent nature of the PB phase produces antisymmetric (equal and opposite) response properties between orthogonal CP states, which means that the same functionality cannot be achieved under right- and left-handed circular polarization (RHCP and LHCP). To overcome this limitation, Guan et al. [82] proposed a polarization-free encoded metasurface to manipulate the circular polarization. The proposed design not only overcame the antisymmetric response properties between orthogonal circular polarization states, thus enabling the same functionality under the illumination of right- and left-handed circularly polarized waves and avoiding polarization transition loss, but also provided additional degrees of freedom for controlling inertia. Guan et al. designed a polarization-free multibit-encoded metasurface for realizing a helical-switching hologram in the microwave region, as shown in Figure 7c. 

In addition, PB phase methods can be combined with other modulation methods. In 2020, Deng et al. [83] presented a multi-freedom metasurface that could simultaneously modulate the phase, polarization, and amplitude independently and, further, realized frequency multiplexing through k-space engineering techniques. The multi-freedom metasurface seamlessly combined geometric Pancharatnam–Berry phases and meander phases, both of which were frequency independent. Thus, it allowed complex amplitude vector holograms at different frequencies based on the same design strategy without the need for complex nanostructure searches for a large number of geometric parameters. Based on this principle, Deng et al. demonstrated visible light full-color complex amplitude metasurface holograms, as shown in Figure 7d.

## 4. Reconfigurable Metasurface Holography

In fact, most of the reported metasurface holography is either static or realizes several different states by using the multiplexing method described above. Recently, reconfigurable metasurfaces have been proposed to provide the possibility of realizing arbitrary, real-time dynamic metasurface holography [84,85]. Reconfigurable metasurfaces integrated with various functional materials (e.g., phase-change materials [86,87], 2D materials [88,89], electronic components [90,91], etc.) allow pixel-level independent control of the optical properties for dynamic metasurface holography through various modulation methods (e.g., thermal excitation, voltage bias, mechanical deformation, etc.) [92,93]. GeSbTe(GST) [94] is a phase-change material widely used in optical storage and reconfigurable photonic devices. It can be repeatedly switched between amorphous and crystalline states thermally, exhibiting different refractive indices and high contrasts in the near- and mid-infrared spectral ranges. By combining a plasmonic metasurface with GST, Zhang et al. [95] realized switchable metasurface holography, as shown in Figure 8a. When the GST was in the amorphous state, the holographic images and vortex beams were performed, as shown in Figure 8b. When heating these devices, the GST changed to a crystalline state, and these functions disappeared. Similarly, liquid crystal [77] is a birefringent material that can be tuned by applying an external electric field or increasing the operating temperature. Rocco et al. [96,97] investigated the directionality of the emitted second harmonic signal generated in a dielectric metasurface consisting of AlGaAs nanocylinders embedded into a liquid crystal matrix, which opened up important opportunities for tunable metadevices such as nonlinear holograms and dynamic displays.

However, it is similar to multiplexed metasurfaces, which can only perform several functions. Loading electronic components to achieve a reconfigurable metasurface is a smarter solution. In 2017, Cui et al. [98] designed a 1-bit digital metasurface loaded with a PIN diode. The digital metasurface had both “on” and “off” scattering characteristics by varying the bias voltage on the PIN diode, as shown in Figure 8c. Therefore, various wavefronts could be dynamically manipulated by controlling the state of the meta-atom with a field-programmable gate array (FPGA). Cui et al. demonstrated an efficient active metasurface hologram by using this method, as shown in Figure 8d. 

In addition, electrically tunable metasurfaces with low latency and field programmability are extremely suitable for combining with deep learning [99,100,101], and there have been some intelligent dynamic holograms by this method capable of instantaneously generating arbitrary targets. In 2021, Liu et al. [41] proposed an unsupervised generative adversarial network physically assisted, as shown in Figure 8e. This network combined the physical mechanism between the electric field distribution and the metasurface and was able to efficiently and noniteratively design encoded metasurface holograms. Liu et al. demonstrated the quick dynamic imaging effect of this method, as shown in Figure 8f. In fact, deep learning has more powerful potential for hologram generation. In 2022, Zou et al. [42] considered the effect of the imaging distance on the physical model and proposed a deep learning network that could generate holograms at the corresponding imaging distance on demand, as shown in Figure 8g. Combined with a reconfigurable metasurface by a varactor diode, Zou et al. demonstrated a 3D hologram slice display, as shown in Figure 8h.

## 5. Discussion and Outlook

Here, we have introduced the development of metasurface holography, from single to multiplexed and reconfigurable. With special nanostructure array designs, metasurfaces can multiplex different optical information into different channels to achieve concurrent holographic tasks with different wavelengths, incident angles, and polarization. From the above discussion, we can find that multiplexed metasurface holography has many applications in optical encryption, security, optical storage, and holographic displays. On the other hand, reconfigurable metasurfaces are not limited to the multiplexing of a finite number of specific wavefronts but are subject to arbitrary on-demand wavefront manipulations through temperature, mechanical, and many other methods of unit control.

Recently, there have been some advances in the research on dynamic video displays with metasurface holography, including space position [102], OAM [67], etc. However, these methods are still far from achieving the ultimate holographic display as shown in science fiction films. Theoretically, an ideal and versatile method to achieve smooth holographic video displays is to control the interaction between the wave and each nanostructure of the metasurface at high speeds, similar to the arrays of LED pixels that display 2D images in our daily lives. Electrically tunable hypersurfaces with field editability are ideal for meeting the above hologram rapid encoding needs. On the other hand, unlike traditional iterative CGH algorithms that are highly time-consuming, data-driven machine learning algorithms have millisecond responses to generate holograms. The excellent performance of the combination of electrically tunable metasurfaces and machine learning in terms of the frame number and frame rate provides a promising alternative method for realizing holographic video displays in specific application scenarios. With the rapid development of nanofabrication and creative CGH algorithms, we believe that ideal holographic video displays will appear in the future.

## Figures and Tables

**Figure 1 nanomaterials-14-00066-f001:**
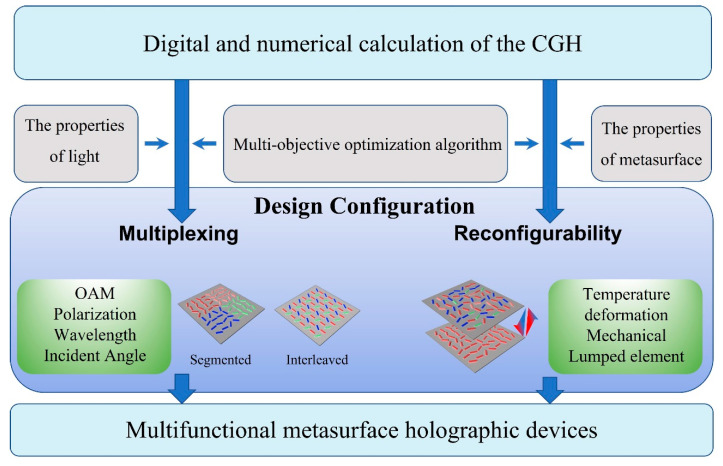
Overview of the procedures for the design of multiplexing and reconfigurable metasurface holography. Important steps include the selection of appropriate multi-objective optimization algorithms and encoding methods. Multiplexing is generally aimed at different optical channels, such as OAM, polarization, wavelength, and incident angle. The configuration of the multiplexing metasurfaces includes segmented, interleaved, etc. Reconfigurability focuses on the variations of the structural parameters that affect the local optical response, and the modulation methods include temperature, deformation, and mechanical and lumped elements.

**Figure 3 nanomaterials-14-00066-f003:**
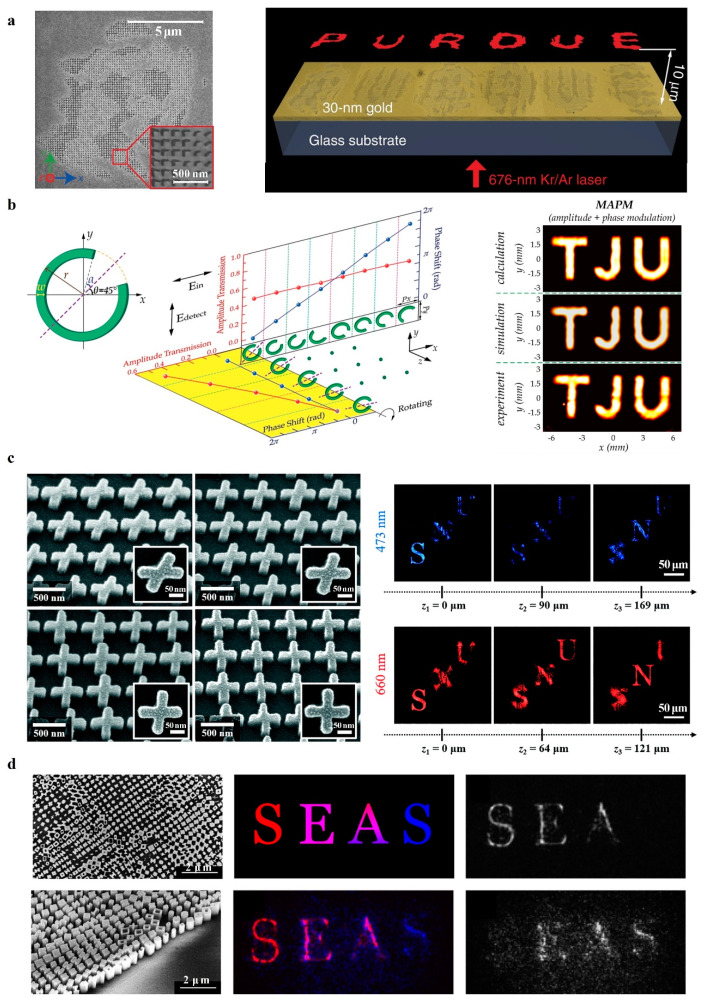
Complex amplitude metasurface holography. (**a**) The V-shaped meta-atom and schematic of metasurface holograms for visible light [37]. (**b**) The CSRR meta-atom and schematic of broadband metasurface holograms: toward the complete phase and amplitude engineering [54]. (**c**) The X-shaped meta-atom and schematic of the complete amplitude and phase control of light using broadband holographic metasurfaces [55]. (**d**) The X-shaped meta-atom and schematic of dielectric metasurfaces for the complete and independent control of the optical amplitude and phase [56].

**Figure 5 nanomaterials-14-00066-f005:**
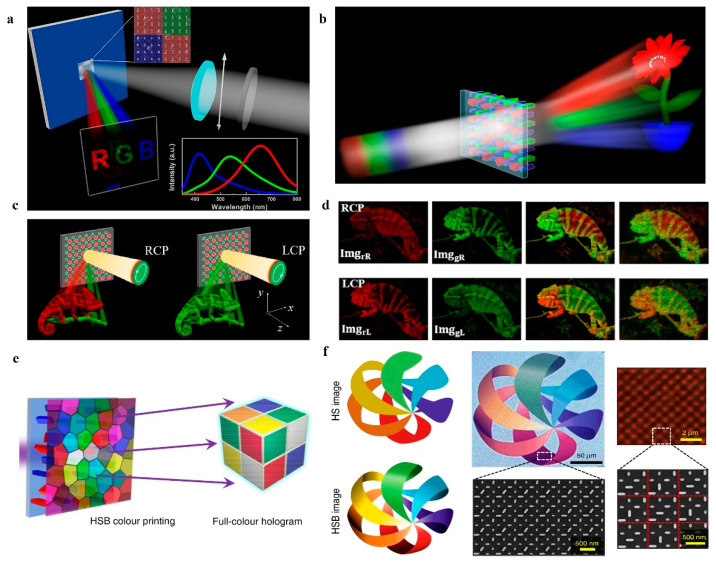
Wavelength multiplexed (colorful) metasurface holography. (**a**) Illustration of a multicolor hologram under linearly polarized incidence in an aluminum nanorod-based array [70]. (**b**) Illustration of a multiwavelength hologram in a dielectric interleaved array [71]. (**c**) Schematic of a polarization-controlled color hologram in a dielectric interleaved metasurface [39]. (**d**) Target holographic images for different polarization states and the corresponding experimental measured results, while lasers of 632.8 and 532 nm provide illumination simultaneously [39]. (**e**) Metasurface for submicron resolution HSB color printing and full-color hologram integration [40]. (**f**) Comparison between HS and HSB images; due to the lack of a brightness dimension, the HS image cannot display the chiaroscuro information [40].

**Figure 6 nanomaterials-14-00066-f006:**
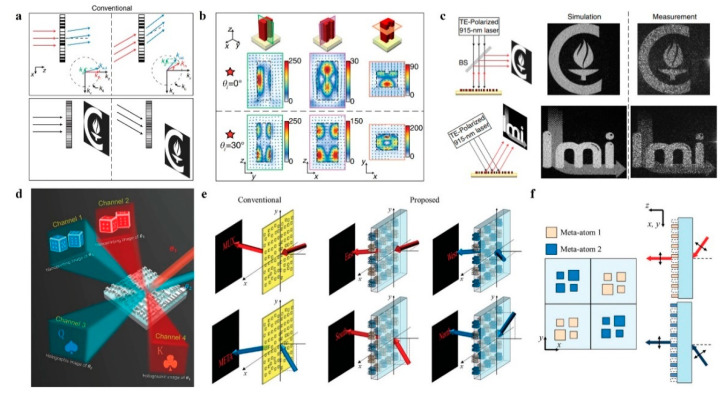
Wavelength multiplexed (colorful) metasurface holography. (**a**) Schematic illustration of the diffraction of light by grating at different angles. (**b**) Different field distributions at normal and 30° incidence are an indication of the excitation of different resonant modes under different incident angles [72]. (**c**) Simulated and experimental measured holographic images captured under a 915 nm TE-polarized laser at 0° and 30° incidence angles [72]. (**d**) Schematic of Independent-Encoded Amplitude/Phase Dictionary for Angular Illumination. Different functions are created under different illumination angles [44]. (**e**) Schematic illustration of the functionality of the detour-phase holograms. Functionality of conventional detour-phase holograms using apertures [74]. (**f**) Top view of a composite composed of meta-atoms 1 and 2 and their diffraction characteristics depending on the incident angles [74].

**Figure 7 nanomaterials-14-00066-f007:**
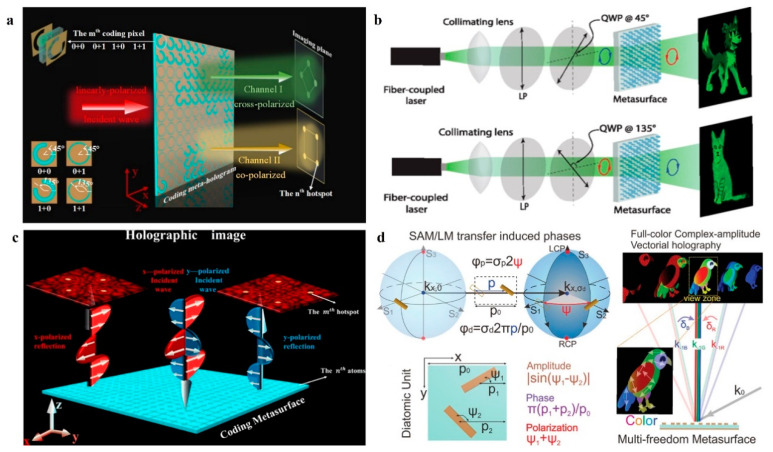
Polarization multiplexed metasurface holography. (**a**) Schematic illustration of dual-polarized multiplexed meta-holograms utilizing coding metasurfaces [78]. (**b**) Schematic diagram and experimental realization of a cartoon dog and cat with tailored Si nanofins for orthogonal circular polarization multiplexing [79]. (**c**) A co-polarization reflection coded hologram under the incidence of *x*- and *y*-polarized plane waves [82]. (**d**) Schematic of a multi-freedom metasurface achieving a full-color complex-amplitude vectorial meta-hologram [83].

**Figure 8 nanomaterials-14-00066-f008:**
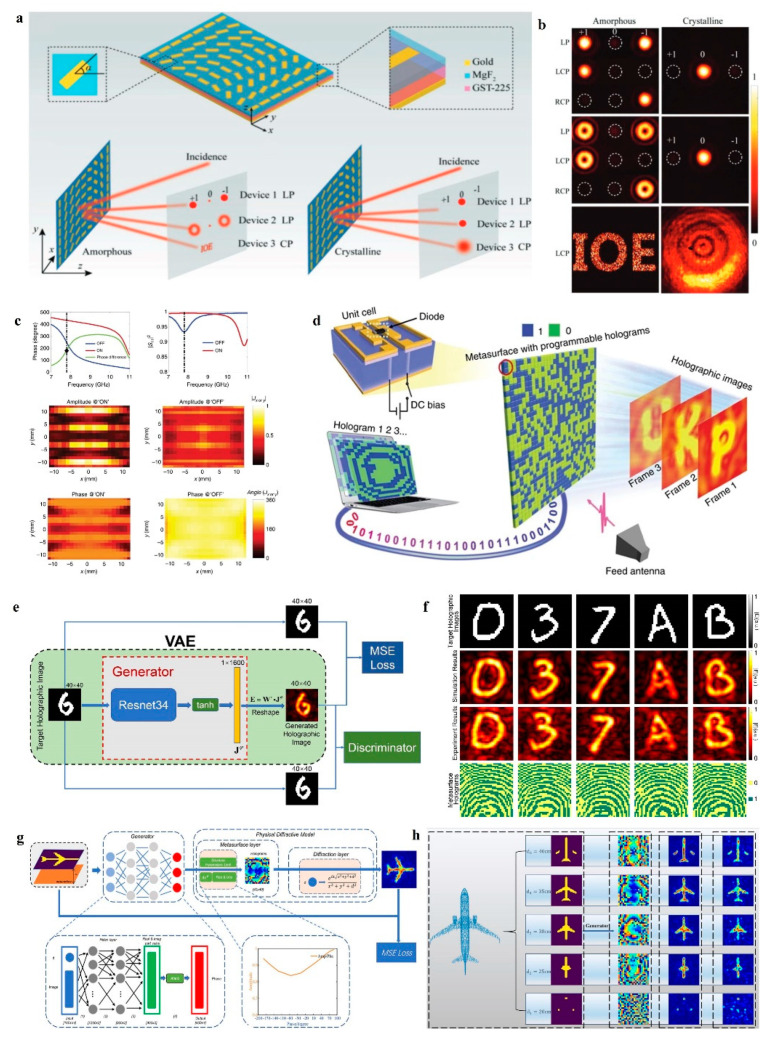
Reconfigurable metasurface holography. (**a**) Schematic of the demonstrated switchable photonic SOIs [95]. (**b**) Different optical performances of three designed meta-devices can be switched when the GST layer is in an amorphous or crystalline state [95]. (**c**) The phase and amplitude of the digital metasurface at the states of “off” and “on” [98]. (**d**) Sketch of the proposed dynamic holographic imaging [98]. (**e**) Schematic diagram of the unsupervised generative adversarial network physically assisted [41]. (**f**) Testing results of the intelligent metasurface hologram system [41]. (**g**) Schematic diagram of the hologram-generating neural networks for the dynamic imaging distance [42]. (**h**) Simulation results of the dynamic imaging distance metasurface hologram system [42].

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
