# Peer review of "Metasurface Holography with Multiplexing and Reconfigurability"

_nanomaterials, 2023, doi:10.3390/nano14010066_

Round 1

Reviewer 1 Report

Comments and Suggestions for Authors

The paper offers a good review of the area of metasurface holography and the variety of meta-elements that have been demonstrated in the last several years. It also reviews potential applications of these elements. 

The structure of the paper is good and the english does not need improvement. 

I recommend publishing the paper in its current state. 

Author Response

Dear Reviewers:

Thank you for your letter and for the reviewers’ comments concerning our manuscript entitled “Metasurface holography with multiplexing and reconfigurability” (ID: nanomaterials-2772662). Those comments are all valuable and very helpful for revising and improving our paper, as well as the important guiding significance to our researches. We have studied comments carefully and have made correction which we hope meet with approval. Please see the attachment for responses to comments.

Reviewer 2 Report

Comments and Suggestions for Authors

Zou et al. have presented a review paper on metasurface holography with specific importance on multiplexing and reconfigurability. The author did a good job by including broad verity of work on multiplexing and reconfigurability. However, the organization of the review has not been up to the mark. Both multiplexing and reconfigurability are broader concept and can be very complex to understand for a general audience. They should include a schematic or some cartoon figure explaining these concepts separately and should emphasize the key things one need to consider to design a metasurface with abovementioned characteristic features. The review should emphasize how to design metasurface with dual characteristic like multiplexing and reconfigurability rather than just discussing whatever is already published. 

Additionally, I have some more specific technical and non-technical comments as listed below.

1.      In line 14-15, the author wrote “conventional pure-structured metasurface holographic devices face a limitation: once fabricated, their functionality remains fixed, restricting practical applications”. I don’t think this is all true. The author should modify the sentence.

2.     The abstract is not up to the mark. The author should emphasize what is the key message of this current review and how it is different than other review available on multiplexing and reconfigurability.

3.     In line 49-50, the author has mentioned metasurface has advantages over metamaterial on “lower fabrication difficulty”. This is not true. The author should modify the sentence.

4.     Line 57 and 58 needs citations to support the claim. 

5.     Figure 2 and other subsequent figure lacking scale bar. The author should discuss about the scale bar in the figure.

6.     In line 227, Fabry-Platino should be replaced with Fabry-Perot.

7.     The summary and discussion haven’t been up to the mark and hence should be re-written. Especially the author should give a more detail overview of their current review. 

Comments on the Quality of English Language

Minor editing is needed.

Author Response

(The authors gave the same response as above.)

Reviewer 3 Report

Comments and Suggestions for Authors

In the paper 'Metasurface holography with multiplexing and reconfigurability' the authors propose a review paper concerning the hot topic of reconfigurable metasurface for holography applications. The paper is generally well written and organized. However, I have few minor comments which I list here:

1. I think the authors should mention all the relevant works related to nonlinear metasurface and holography. For instance they completely neglect the works related to the shaping of the SH emission pattern of a dielectric nanoantenna by integrated holographic gratings 

2. At the same time, I think that the authors should mention a lot of relevant works concering the reconfigurability concept in the linear and nonlinear regime. In the present form of the manuscript a lot of recent works are missing. More specifically, I am referring to the field of opto-thermally beam steering in nonlinear all-dielectric metastructures, tunable SHG by all-dielectric diffractive metasurface embedded in liquid crystals and in general the switching of the nonlinear signal by all dielectric metasurface via tunable liquid crystal. 

3) When talking about 'phase-only metasurfaces' the authors should mention the concept of metalens which is of dramatically importance for different research fields.

Author Response

(The authors gave the same response as above.)

Round 2

Reviewer 2 Report

Comments and Suggestions for Authors

The author addressed my comments and suggestions. I have no additional comments.

Comments on the Quality of English Language

Minor changes needed.